# Mapping of histone-binding sites in histone replacement-completed spermatozoa

Keisuke Yoshida [1], Masafumi Muratani [2], Hiromitsu Araki[3], Fumihito Miura[3], Takehiro Suzuki[4], Naoshi Dohmae[4], Yuki Katou[5], Katsuhiko Shirahige[5], Takashi Ito[3] & Shunsuke Ishii[1,6]

The majority of histones are replaced by protamines during spermatogenesis, but small amounts are retained in mammalian spermatozoa. Since nucleosomes in spermatozoa influence epigenetic inheritance, it is important to know how histones are distributed in the sperm genome. Conflicting data, which may result from different conditions used for micrococcal nuclease (MNase) digestion, have been reported: retention of nucleosomes at either gene promoter regions or within distal gene-poor regions. Here, we find that the swim-up sperm used in many studies contain about 10% population of sperm which have not yet completed the histone-to-protamine replacement. We develop a method to purify histone replacement-completed sperm (HRCS) and to completely solubilize histones from cross-linked HRCS without MNase digestion. Our results indicate that histones are retained at specific promoter regions in HRCS. This method allows the study of epigenetic status in mature sperm.

[1] Cluster for Pioneering Research, CREST Research Project of JST (Japan Science and Technology Agency), RIKEN Tsukuba Institute, Tsukuba, Ibaraki 305-0074, Japan. [2] Department of Genome Biology, Faculty of Medicine, University of Tsukuba, Tsukuba, Ibaraki 305-8575, Japan. [3] Department of Biochemistry, Kyushu University Graduate School of Medical Sciences, Fukuoka 812-8582, Japan. [4] Biomolecular Characterization Unit, RIKEN Center for Sustainable Resource Science, Wako, Saitama 351–0198, Japan. [5] Institute for Quantitative Biosciences, The University of Tokyo, Tokyo 113-0032, Japan. [6] Department of Functional Genomics, Graduate School of Comprehensive Human Sciences, University of Tsukuba, Tsukuba, Ibaraki 305-8575, Japan. Correspondence and requests for materials should be addressed to K.Y. (email: Keisuke.Yoshida@riken.jp) or to S.I. (email: sishii@rtc.riken.jp)

Increasing evidences indicate that various paternal environmental factors affect the gene expression pattern and traits in an offspring. Paternal nutritional conditions such as a high-fat or low-protein diet induce gene expression changes in the pancreatic islets of rat or the liver of mouse offspring, respectively[1,2]. Furthermore, in utero undernourishment perturbs the metabolism in offspring via sperm[3]. Although the mechanism of paternal inheritance of environmentally induced changes remains elusive, some evidences suggested that histone modification changes play an important role[4]. We previously demonstrated that *Drosophila* transcription factor ATF2 mediates the heat shock-induced or osmotic stress-induced reduction of histone H3 lysine 9 trimethylation (H3K9me3), which is transmitted to the next generation[5]. The other group also reported that overexpression of the histone H3 lysine 4 (H3K4) demethylase KDM1A (also known as LSD1) during spermatogenesis impairs the development and survivability of the offspring[6]. However, the molecular mechanisms underlying paternal epigenetic transmission are still unclear.

To understand these mechanisms, it is important to know the distribution of histones in mature sperm. During mammalian spermatogenesis, most histones are replaced by protamines in spermatids[7]. Histone-to-protamine replacement is essential for inducing sperm nuclear condensation, which is one of the most important parameters for estimating human sperm quality. Indeed, round spermatid nucleus injection into oocytes has a lower fertilization rate than intracytoplasmic sperm injection[8]. In the clinical field of assisted reproduction techniques, mature sperm have traditionally been prepared from human semen mainly using two methods: Percoll gradient centrifugation or the swim-up procedure[9]. The Percoll procedure to isolate normal sperm were the first innovative method to improve pregnancy rates in human in vitro fertilization. The swim-up method is now popularly used for isolating high-motility sperm and removing the abnormal sperm. Some studies suggest that the Percoll method is more effective than the swim-up method in producing samples with higher nuclear sperm condensation and lower populations of morphologically abnormal sperm[10,11]. These results suggest the importance of sperm nuclear structure for fertility.

The sperm chromatin structure assay (SCSA) was developed for diagnostic assessment of human sperm quality by analyzing the nuclear structure of the sperm[12]. This assay uses acid treatment to denature DNA at the sites of DNA strand breaks, followed by staining with acridine orange (AO). AO intercalates mainly into double-stranded DNA containing histones, but not protamines, which then generates green fluorescence (Supplementary Fig. 1a, b). On the other hand, AO stacks on single-stranded DNA, which is formed by apoptosis-induced DNA double-strand breaks, generating red fluorescence. Thus, the fraction of sperm with high DNA stainability (HDS), as indicated by green fluorescence, has an abnormally high level of DNA staining due to lack of full protamination and consists of sperm population with incomplete histone replacement[12]. Actually, the group with a larger proportion of HDS sperm in humans has a lower chance of pregnancy[13,14]. The DNA fragmentation index (DFI), as measured by red fluorescence, indicates DNA fragmentation caused by apoptosis. The content of DFI fraction in mouse sperm samples is usually lower than in human samples, possibly because mouse sperm can be analyzed immediately after preparation and has less histones compared with human sperm.

A small fraction of histones remains associated with the sperm genome[15,16]. By deep sequencing of chromatin DNA immunoprecipitated with anti-histone antibodies from mononucleosomes solubilized by micrococcal nuclease (MNase) without cross-linking, Hammoud et al. reported that the retained nucleosomes in human sperm are enriched at development-related genes, including imprinted gene clusters and the promoters of developmental transcription and signaling factors[15]. Using a similar method, Erkek et al. also showed that mouse spermatozoa contain only 1% of residual histones, which are ten- and twofold more enriched at GC-rich promoters and exons, respectively, relative to genomic background[16].

However, two groups recently reported data inconsistent with those of the previous reports. By high-throughput sequencing of the mononucleosomal DNA released by MNase, Samans et al. showed that most nucleosomes from human and bovine sperm are enriched in distal intergenic regions and localized in repetitive DNA sequences[17]. Using a recently developed nucleosome mapping method, Carone et al. also reported that the nucleosomes in mouse sperm are preferentially retained in gene-poor regions and are generally depleted from promoters, including developmental promoters such as the Hox promoters[18]. Carone et al. speculated that such discrepancies might result from differences in the MNase digestion conditions of the isolated sperm chromatin because they recovered mononucleosomes enriched over promoters of developmental regulators when a relatively high concentration of MNase was used. However, Samans et al.[17] used relatively high concentrations of MNase similar to those used by Hammoud et al., but obtained contradictory results[15,17]. Furthermore, the nucleosomal enrichments at repetitive sequences reported by Samans et al.[17] resulted from the redundant use of sequencing reads that map to multiple locations in the genome, an inappropriate computational methodology[19,20]. Thus, two inconsistent results have been reported[21], and a resolution of the conflicting data is needed for understanding the epigenetic status in sperm genome.

Here, we carefully examine the quality of sperm and find that the swim-up sperm fraction used by most research groups still contains the histone replacement-uncompleted sperm (HRunCS). By developing a method to purify the histone replacement-completed sperm (HRCS) and to completely solubilize histones after cross-linking without MNase digestion, we demonstrate that histones are distributed on the promoters of specific genes in HRCS. Results obtained by our method suggest that histones in mouse spermatozoa may contribute to expressional regulation of the target genes during early embryonic development.

## Results

**Isolation of the HRCS.** To examine the quality of sperm, we first analyzed the sperm using SCSA[12]. Spermatozoa develop from spermatogonia stem cells in the testis and then move to the epididymis, and into the vas deferens (Fig. 1a). To check the quality of the mouse sperm, we performed SCSAs for total sperm fraction collected from different parts of the epididymis or vas deferens (Supplementary Fig. 2a, b). We found the HDS fractions of 26.8% and 20.3% in the total sperm from caput and corpus epididymis, respectively (Fig. 1a). HDS fractions of 11.6% and 8.7% were observed in the total sperm from the cauda epididymis and the vas deferens, respectively. These results indicated that the fraction of HDS, which has not completed the histone-to-protamine replacement, decreases during maturation in the epididymis, whereas the fraction of normal sperm, which has completed histone replacement, increases. Consistent with this, western blotting results showed that the amount of histone H3 decreased during movement from caput to cauda epididymis (Fig. 1b). These results suggest that the histone replacement by protamine is not completely finished when the sperm enters the epididymis, and is gradually completed during the movement of sperm from the caput to the cauda of the epididymis. Alternatively, the histone replacement-incomplete sperm might be removed by apoptosis during epididymal maturation. In addition,

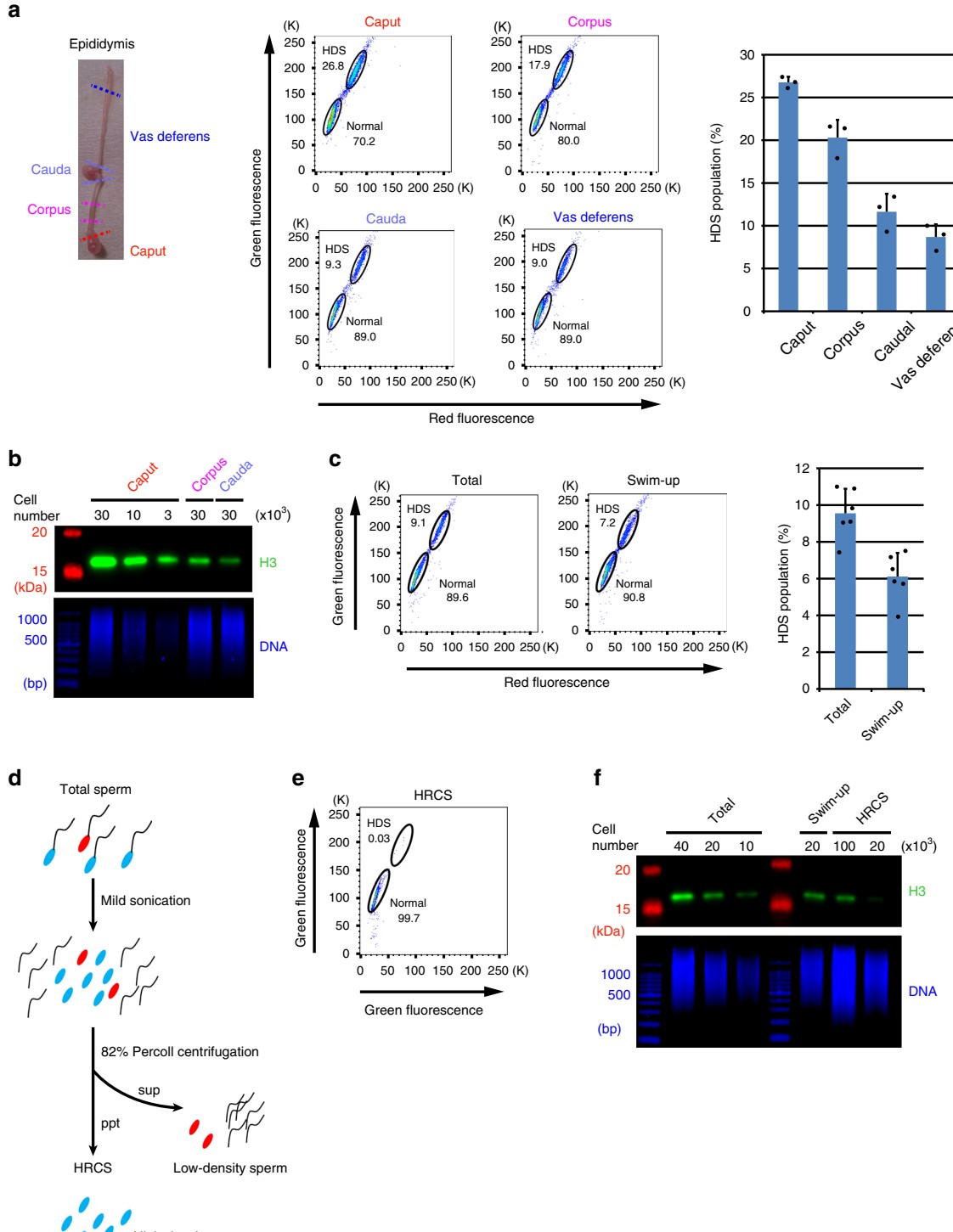

**Fig. 1** Estimation of histone amount in each sperm fraction. **a** (Left) Picture of mouse epididymis consisting of caput, corpus, caudal epididymis, and vas deferens. (Middle) SCSA results for total sperm fractions collected from each part of the epididymis and vas deferens. Only cell population of singlet is shown here. Sperm cells are mainly separated into two fractions: normal sperm (normal) and the histone replacement-incomplete sperm with high DNA stainability (HDS) with AO. (right) The mean ± s.d. of percentage of HDS population ($n = 3$). **b** Immunoblot analysis of histone H3 in total lysates of sperm fraction collected from each part of the epididymis. Cell numbers applied to each lane are indicated by thousands. DNA was analyzed for verifying the cell number of each sample. **c** (Left) SCSA results for total sperm and swim-up sperm fraction. (Right) The mean ± s.d. of percentage of HDS population ($n = 6$). **d** Schematic procedure for preparation of the histone replacement-completed sperm (HRCS). Sperm cells with red or blue heads indicate HRunCS or mature sperm, respectively. High-density sperm heads generated by mild sonication were isolated by Percoll. Note that some portion of HRunCS could be sensitive to mild sonication. **e** SCSA results for HRCS. **f** Immunoblot analysis of H3 in total sperm, swim-up sperm, and HRCS preparation

the sperm from cauda epididymis and the vas deferens still contain significant amounts of HRunCS.

To examine the histone distribution in sperm chromatin, a relatively large amount of sperm is required, and the amount from the vas deferens is not sufficient. Consequently, in most of the experiments reported so far, sperm from the cauda epididymis, which contains ~tenfold more sperm than the vas deferens, were used. Furthermore, sperm with high motility selected by the swim-up method (swim-up sperm) were used in most of the studies. Therefore, we next analyzed the swim-up sperm using SCSAs. Unexpectedly, 6.1% of the swim-up sperm preparation showed HDS (Fig. 1c). To test whether the population of HDS varies depending on the procedure of the swim-up method used, the top 0.8 ml medium of 4.0-ml caudal epididymis cultivation (2.5 ml/4.0 ml in the standard procedure[22]) was also recovered and analyzed. About 8.7% of the prepared sperm showed HDS (Supplementary Fig. 2c), indicating that the population of HRunCS does not vary depending on the procedure used.

To avoid contamination by histones derived from HRunCS for mapping of H3-binding sites, we developed a method to isolate the HRCS from total sperm collected from caudal epididymis and vas deferens (in the following description, total sperm means total sperm fraction prepared from caudal epididymis and vas deferens), based on Percoll centrifugation for purification of high-density sperm heads. The total sperm were mildly sonicated to generate sperm heads by removing the tails and centrifuged in 82% Percoll solution. The high-density sperm were spun down, while the relatively low-density sperm and sperm tails remained in the supernatant (Fig. 1d and Supplementary Fig. 2d). SCSAs of the pelleted sperm indicated that this preparation contained almost no HDS fraction (Fig. 1e), and almost 100% purity of the HRCS fraction was obtained. The H3 antibody used in this study recognized H3.1 and H3.3 with the same degree of sensitivity (Supplementary Fig. 2e). Results of western blotting using this antibody indicated that the amount of histone H3 in the HRCS was about 1/5 that of the total sperm and that of the swim-up sperm (Fig. 1f). Target MS–MS quantification analysis also showed that the amount of histone H3 variants H3.1, H3.2, H3.3, or H3t in the HRCS was about 33, 14, 27, or 29% that of the total sperm, respectively (Supplementary Fig. 3). These results suggest that HRunCS contains much more amount of histone proteins than HRCS. In total sperm, 10% of them was HRunCS, and results of western blotting indicated that the ratio of H3 in total sperm (90% HRCS + 10% HRunCS) and HRCS (100% HRCS) is 5:1. These results indicate that H3 protein contained in total and swim-up sperm samples is mainly derived from HRunCS, and that the distribution of histone-binding sites in the swim-up sperm reflects the localization of histones in HRunCS. Comparison of the histone H3 level by western blotting analysis showed that the amount of H3 in HRCS was about 0.3% of that in somatic cells of mouse embryonic fibroblasts (MEFs) (Supplementary Fig. 1f), which is lower than that previously reported in swim-up sperm[23,24].

**Complete solubilization of histones from cross-linked HRCS.** Digestion of sperm chromatin using a high concentration of MNase might fail to detect specific regions of chromatin that are hypersensitive to MNase digestion. We observed that the tiny amount of nucleosomes contained in HRCS could not be completely solubilized even using a high concentration of MNase or sonication after cross-linking. Therefore, to avoid the effect of MNase digestion conditions, we developed a method to completely solubilize the nucleosomes of sperm without MNase digestion after cross-linking. Compared with the cross-linked

somatic cells, cross-linked sperm are very hard and it is difficult to disrupt and solubilize nucleosomes by sonication. For these reasons, we developed a buffer to induce decondensation of the cross-linked sperm nuclei. Protamine is a strong basic protein and tightly binds to sperm DNA via ionic interactions between the positive charge on arginine residues of protamine and the negative charge on phosphate groups of DNA[25]. Furthermore, disulfide linkage between protamine molecules is required for compaction of sperm DNA[26]. We therefore treated sperm cells with dithiothreitol (DTT) for reducing the disulfide bonds and with heparin for neutralizing the positive charge of protamines (Fig. 2a). Heparin induces decondensation of sperm chromatin[27]. After this treatment, sperm heads were enlarged (Fig. 2b), indicating decondensation of sperm nuclei. When these cells were treated by sonication, all the sperm DNA and nucleosome histone H3/H4 were recovered in the supernatant after centrifugation (Fig. 2c), indicating that most of the nucleosomes were solubilized, and DNA was sheared to the appropriate size for ChIP experiments using this method.

**Binding profiles of histone H3 in total sperm and HRCS.** Using these conditions, we performed histone H3 ChIP-seq analysis in total sperm and HRCS. H3 signals in total sperm and HRCS were mainly found on gene promoter regions such as those of the *Hoxa* gene cluster, and the signals were considerably less intense or absent in HRCS (Fig. 3a and Supplementary Fig. 4). Genome-wide analysis indicated that 10,988 peaks in total sperm and 1320 peaks in HRCS were detected, and this difference in peak number may be consistent with the amount of H3 in total sperm and HRCS. Approximately 69% and 64% of peaks were localized to promoter regions in total sperm and HRCS, respectively (Fig. 3b), and H3 peaks were enriched in promoter regions and exon regions relative to background (Supplementary Fig. 5a). A minor population of peaks was found in intergenic repeat regions such as LINE and GC-rich repeats (Supplementary Fig. 5b). In the peak regions detected in total sperm, peak strength in HRCS was approximately half that in total sperm (Fig. 3c). About 73% of peaks (966/1320) in HRCS overlapped with those in total sperm, while most of the H3 target genes in HRCS (905/911) were also target genes in total sperm (Fig. 3d). About 64% of the H3 peaks (7067/10,988) detected in total sperm overlapped with the nucleosome regions detected by MNase-seq using swim-up sperm in the previous report[16] (Supplementary Fig. 5c), suggesting similar histone-binding profiles in swim-up and total sperm. Consistent with this, both populations contained similar HDS fractions (Fig. 1c). About 22% (285/1320) of histone H3 peaks and 75% (686/911) of H3-binding genes in HRCS overlapped with those in the swim-up sperm, but the number of H3 peaks and H3-binding genes was much lower than in swim-up sperm[16].

Around 76% (8351/10,988) and 67% (879/1320) of H3 peaks in total sperm and HRCS, respectively, were observed in CpG islands (CGIs) (Fig. 3e). The median observed/expected (o/e) ratios in HRCS peaks were higher than those in total sperm peaks and all CGIs, indicating that histone H3 is preferentially retained at CGIs with higher CG dinucleotide enrichment in mature sperm. Interestingly, H3 peaks were not uniformly distributed on each chromosome and were observed at lower frequencies only on sex chromosomes (Supplementary Fig. 5d). This tendency might be explained by the lower frequencies of CGIs with high o/e ratios (>1.0) on sex chromosomes relative to autosomal chromosomes. The DNA methylation profile of HRCS was not much different to that of the swim-up sperm (Supplementary Fig. 6a). Furthermore, H3-binding sites in total sperm and HRCS were predominantly in lower methylation regions, consistent with the previous report[16] (Supplementary Fig. 6b). The DNA

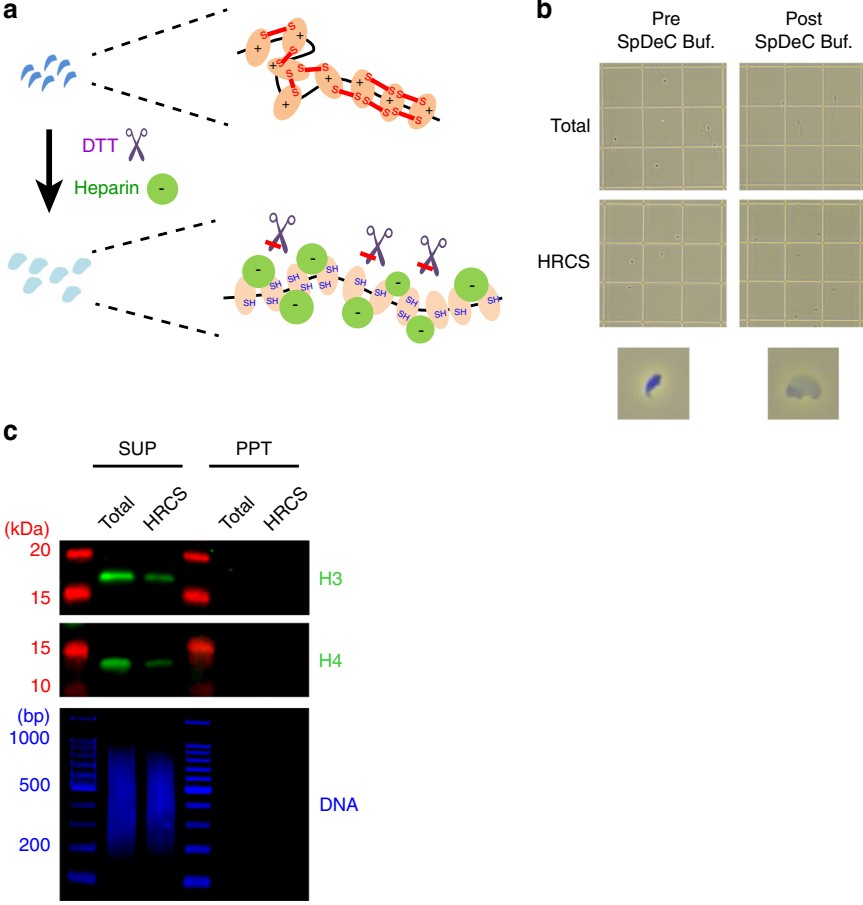

**Fig. 2** Solubilization of histones and DNA from cross-linked sperm cells after decondensation treatment **a** Schematic diagram for expected molecular reaction during sperm decondensation treatment. (Top) Sperm DNA is highly compacted by polymerized protamine via disulfide bonds (red lines), and protamine tightly binds to DNA through ionic interactions. (Bottom) After treatment with sperm decondensation buffer, the reducing capacity of dithiothreitol (DTT) cleaves the disulfide bonds, and the positive charge in protamine molecules is neutralized by the negative charge of heparin, leading to decondensation of the DNA–protamine complex. **b** The morphology of total sperm and HRCS fraction before and after treatment with sperm decondensation buffer. Sperm morphology was observed under a MAKLER-style cell counter (0.1 × 0.1 mm). **c** Total sperm and HRCS fractions were cross-linked, incubated with sperm decondensation buffer, sonicated, and centrifuged. Immunoblot analysis of H3 and H4 using the supernatant and pellet fraction from $4 \times 10^4$ cells was performed

methylation level was not significantly different between these H3-binding sites (Supplementary Fig. 6c).

We further analyzed the histone H3-binding genes, which we divided into two groups, HRCS target genes, where H3 peaks were detected in HRCS, and total sperm-specific target genes, where H3 peaks were detected only in total sperm samples, but not in HRCS. Motif analysis showed that repeated CGG sequence was extracted from binding sequences in HRCS target genes, whereas CC was extracted from those in total sperm-specific target genes (Supplementary Fig. 7a), supporting the observation that the o/e ratio of CpG in the H3-binding sites in HRCS was higher than that in total sperm (Fig. 3e). Using published data[16], we investigated the profiles of histone modifications around the transcription start sites (TSSs) of these H3-binding genes in RS and swim-up sperm. There was no specific difference in H3K4me3 and H3K27me3 in RS and swim-up sperm between HRCS target and total sperm-specific target genes (Supplementary Fig. 7b). Thus, these two target genes are not defined at least by H3K4me3 or H3K27me3 marks.

**Epigenetic feature of sperm H3-binding genes in the embryo.** The heatmap data of H3 signal showed that the HRCS target

genes had a large amount of H3 also in the total sperm (Fig. 4a). A group of genes in total sperm-specific target genes also had a moderate level of H3 for the HRCS data. Therefore, we further divided the H3-binding genes into three categories: category_H corresponded to HRCS target genes, category_PH (potential HRCS target genes) to total sperm-specific target genes with moderate levels of H3 in the HRCS data, and category_TS (total sperm-specific target genes) to total sperm-specific target genes (Fig. 4a). Pathway analysis indicated that category_H genes included a group of genes related to developmental biology, which include the genes encoding Hox transcription factors and also related to neuronal cell differentiation (axon guidance and signaling by NGF), while category_TS involved genes related to gene expression and metabolism of proteins (Fig. 4b and Supplementary Data 1). Only development-related genes were sheared between category_H genes and nucleosome target genes previously identified by MNase-seq[16] (Supplementary Fig. 8a).

Using published data[28,29], we also investigated the chromatin state and epigenetic profiles of these target genes during early embryogenesis because they are important for understanding molecular mechanisms of epigenetic inheritance via sperm. At the four-cell stage, category_H and category_PH genes showed

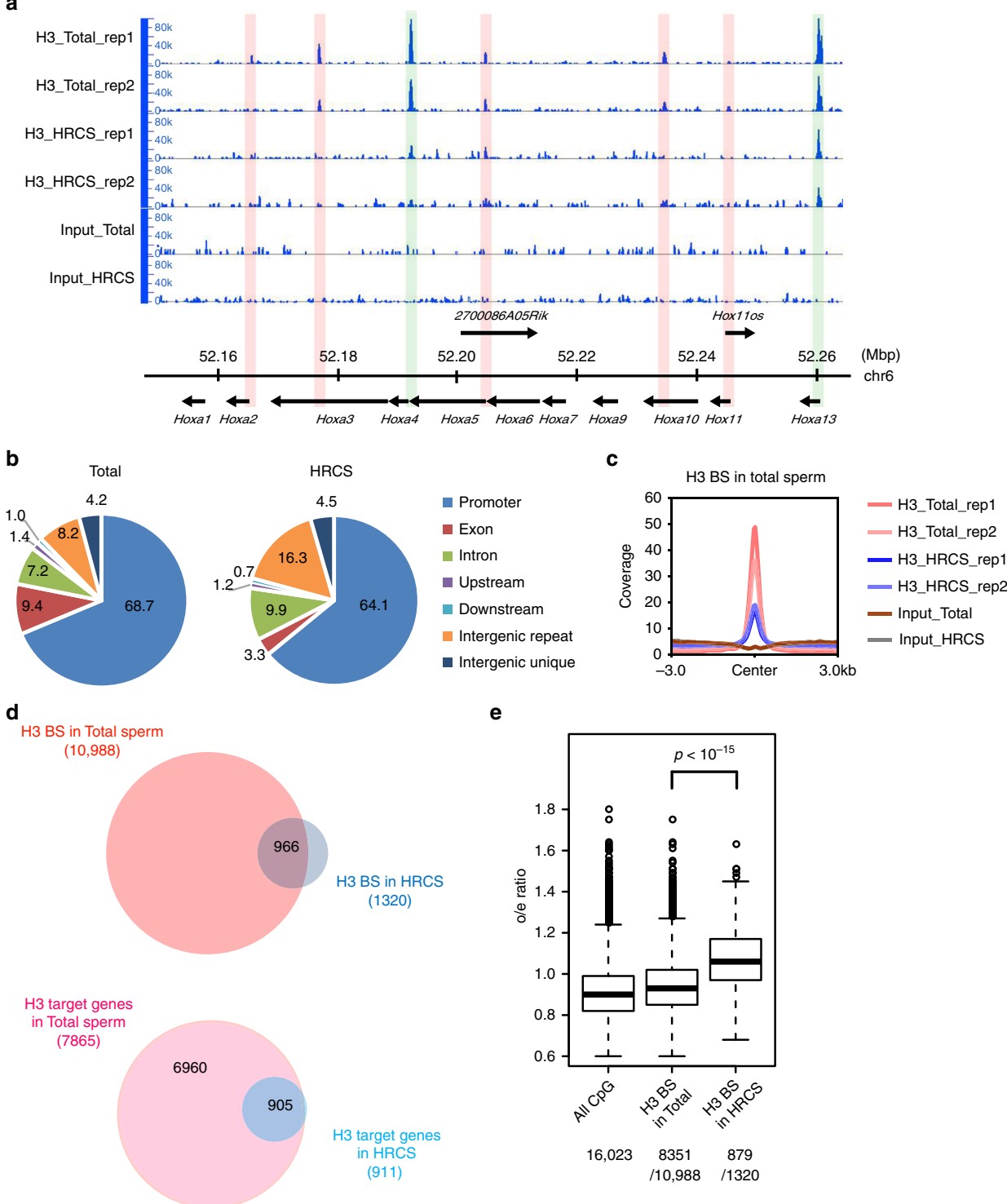

**Fig. 3** Analysis for H3-binding regions in total sperm and HRCS. **a** H3 ChIP-seq analysis for total sperm and HRCS in *Hoxa* gene cluster regions. Blue signal indicates the number of raw reads, which is scaled to mapped regions in each sample. Light blue boxes indicate the binding sites detected in both total sperm and HRCS. Light red boxes indicate the binding sites detected only in total sperm. **b** Distribution of H3 peaks detected in total sperm and HRCS (10,988 and 1320 peaks, respectively) among various genomic regions. **c** Profile plot for signal intensity of H3 ChIP-seq around binding sites detected in total sperm and HRCS. **d** Overlap of H3-binding sites (top) or H3 target genes (bottom) between total sperm and HRCS. **e** Box plot for o/e ratio in all CpG islands or the CpG islands overlapping with H3-binding sites in total sperm or HRCS. The number of CpG islands overlapping with H3-binding sites in total sperm or HRCS is shown at the bottom. The elements in box plot indicate following value; center line, median; box limits, upper and lower quartiles 1.5x interquartile range; and points, outliers. P value was calculated by Wilcoxon rank-sum test

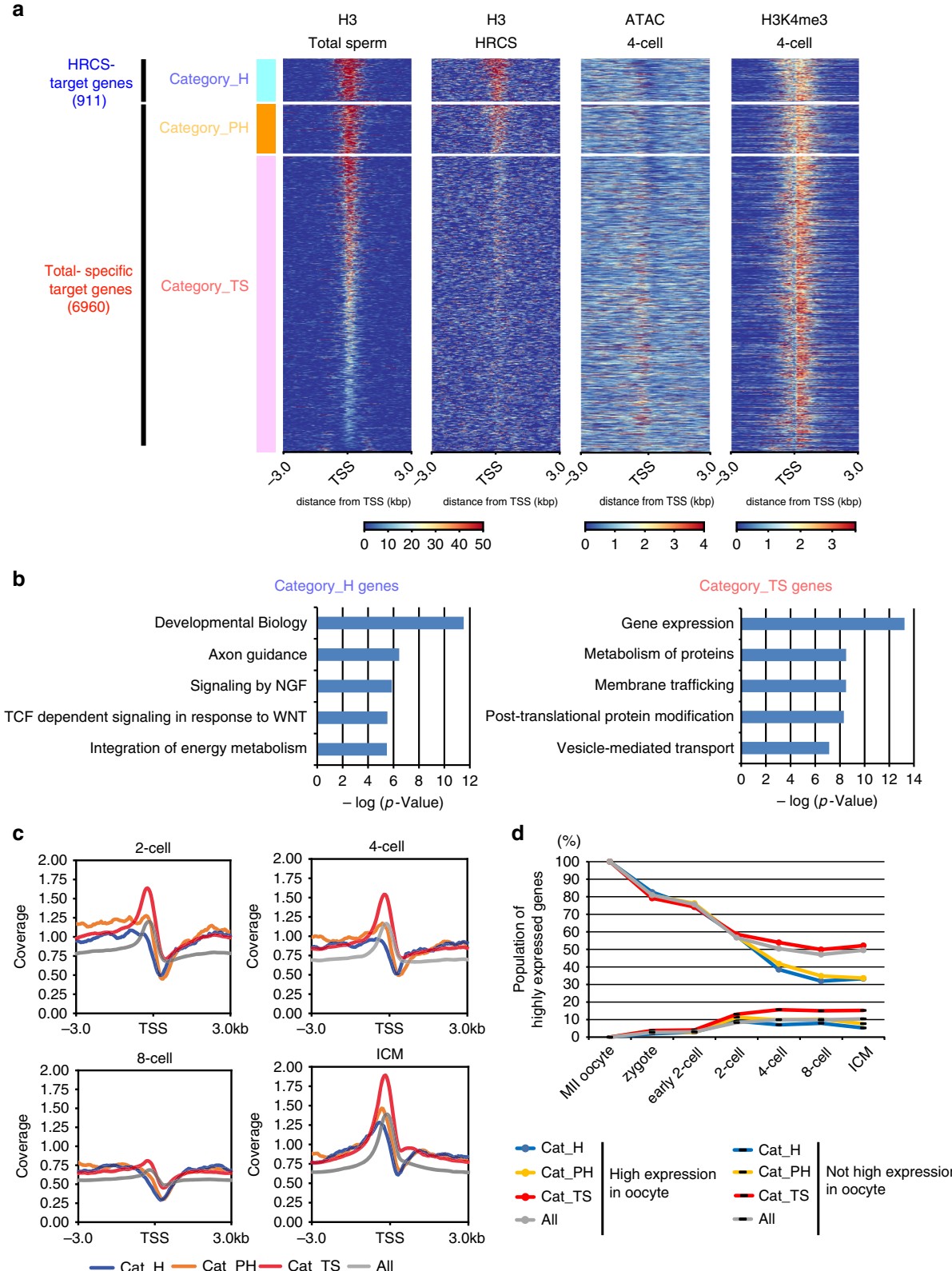

lower ATAC-seq and H3K4me3 ChIP-seq signals than category_TS genes, suggesting that the chromatin structure of category_H/PH genes exhibits lower DNA accessibility and slightly weaker transcriptional activity than that of category_TS genes at this stage (Fig. 4a) (Supplementary Fig. 8b and Supplementary Fig. 9). The profile plot of ATAC-seq signal also showed a sharp drop around the TSS of category_H/PH from the two-cell to eight-cell stage, but a clear peak around the TSS of category_TS and all genes from the two-cell to four-cell stage (Fig. 4c).

**Fig. 4** Contribution of H3 binding in sperm to gene silencing during early embryogenesis. **a** Heatmap of signal intensity of H3 ChIP-seq in total sperm and HRCS, ATAC-seq at the four-cell stage, and H3K4me3 ChIP-seq at the four-cell stage, across H3 target genes in total sperm and HRCS. H3 target genes are categorized into three groups: category_H (target genes in HRCS), category_PH (Potential target genes in HRCS), and category_TS (specific target genes in total sperm), based on the signal strength of the H3 peak in HRCS. **b** Pathway analysis for category_H and category_TS genes using Reactome datasets. The five top pathway names are presented. **c** Profile plot for signal intensity of ATAC-seq around the TSS of all genes and category_H, category_PH, and category_TS genes. **d** Percent population of highly expressed genes (FPKM > 10), which are contained in all genes or category_H, category_PH, or category_TS target genes, from oocyte to ICM. The target genes in sperm and all genes were separated by expression level in the oocyte (FPKM > 10)

Finally, using published RNA-seq data[29], we analyzed the dynamics of expression profiles for the H3-binding genes in HRCS and total sperm during early embryogenesis. Interestingly, the proportion of highly expressed genes in category_H/PH was reduced drastically after the four-cell stage, but that in category_TS and total genes was moderate (Fig. 4d and Supplementary Fig. 10a, b). Consistent with this, H3K4me3 signal in the H3 target genes of HRCS was slightly lower than those of total sperm in early embryos (two-cell to ICM) (Supplementary Fig. 9). These data suggest that the maternal mRNAs of category_H/PH genes are degraded at the two-cell stage, and most of those mRNAs are not induced during zygotic gene activation (ZGA) in the four-cell embryo, while the genes of category_TS are transcriptionally induced during ZGA. To test whether this might be correlated with the presence of epigenetic silencing marks in HRCS, we have examined that these genes have the histone H3K9me2/3 marks. Actually, H3K9me2, not H3K9me3, was found in the promoter regions of category_H genes, whose expressions were suppressed at the stages between four-cell and ICM (Supplementary Fig. 11a, b). Thus, H3 binding and/or modified histone in sperm may silence the target genes during early embryogenesis.

## Discussion

Here, we developed a method to analyze histone distribution in mouse sperm by using two modifications: the purification of histone-to-protamine replacement-completed sperm, HRCS, and the complete solubilization of nucleosomes after cross-linking, using DTT and heparin treatment and sonication without MNase digestion. The results obtained using this method indicate that histones are localized at the promoter regions of specific genes. The data using the swim-up sperm previously reported by Erkek et al.[16] also indicated histone distribution on gene promoter regions, although fewer H3-binding genes were observed in HRCS than in the swim-up sperm. The data obtained using swim-up sperm are consistent with our results using total sperm from the cauda epididymis, which contain a similar population of HRunCS to the swim-up sperm. Our results did not show the enrichment of histone H3 in gene-poor intergenic regions and are therefore inconsistent with the results reported by the two groups[17,18]. The reason for this discrepancy is unclear, but in the present study, histones were cross-linked to DNA before extraction, and the solubilization of histones by sonication was almost complete. In addition, MNase was not used in our method, so the results were not affected by the MNase digestion condition.

Recently, Barral et al. analyzed the process of the histone-to-protamine replacement and showed the nature of transitional structures[30]. In condensing spermatids, the H2A.L.2–TH2B dimer is first loaded onto the nucleosomes, which opens the nucleosomes and allows the invasion of nucleosomes by transition proteins. Nucleosome–transition proteins interact with the incoming protamines, which mediate pre-protamine-2 processing, and the protamines then bind to DNA in competition with histone–transition protein complexes. Since the displaced histones are unable to remain as octamers, protamine–DNA and displaced transition protein–histone complexes constitute a relatively stable transitional state, generating small subnucleosomal structures. HRCS might involve the small subnucleosomal structures, and our sample may contain them because MNase was not used in our method to discriminate the normal nucleosomal structures and the small subnucleosomal structures. Thus, further analyses might be needed to examine whether the small subnucleosomal structures might contribute to some H3-binding genes detected in HRCS.

We found HDS fractions, representing histone replacement-incomplete sperm, of 26.8%, 20.3%, and 11.6% of the total sperm from caput, corpus, and cauda epididymis, respectively. This suggests that sperm have not yet completely finished the histone-to-protamine replacement after leaving the testis, and histone replacement continues during movement in the epididymis. When sperm cells enter the epididymis, they have not yet acquired the capacity to move and are unable to fertilize the oocytes. They need the epididymal maturation process during movement in the epididymis to obtain fertilization capacity. The epididymal maturation involves a series of molecular events including the increased level of sialic acid residue, disulfide bond formation, membrane fluidity changes (caused by a decrease in the cholesterol/phospholipid ratio), phosphorylation of Izumo1, and tRNA fragmentation (see the review by Gervas and Visconti[31] and references therein). However, the completion of histone-to-protamine replacement has not been recognized as an event of epididymal maturation, and the present study indicates that SCSA is a useful method to judge the completion of histone-to-protamine replacement in the study of mouse sperm. Our results also showed that the HDS fraction of the sperm from either cauda epididymis or vas deferens was approximately 10%. This may not be surprising because the HDS fraction of sperm from healthy men is also 5–10%[32]. We have developed a method to purify HRCS using Percoll gradient centrifugation of high-density sperm. The amount of histone H3 in HRCS was about 1/5 that of swim-up and total sperm from cauda epididymis. Consistent with this, H3 binding in HRCS was observed in the promoter regions of fewer genes than in total sperm. These results suggest that preparation of high-density sperm using Percoll centrifugation is a more useful method than the widely used swim-up method, also in the clinical field of assisted reproduction techniques.

H3 binding mainly to the promoter region in HRCS suggests that its role might be transcriptional regulation after fertilization. The expression level of the H3-binding genes in HRCS tends to be high in oocytes and suppressed after the two-cell stage. Although further analysis is required to examine whether residual histones in sperm affect gene expression after fertilization, recent reports suggest that histones in gametes regulate epigenetic status in the fertilized egg[33,34]. The previous study using swim-up sperm indicates that H3.3 is mainly incorporated into sperm chromatin at a higher rate than H3.1/H3.2[16]. H3.3 contributes to gene silencing in ES cells by gain of H3K9me3 via ESET[35,36]. These results suggest that inherited H3.3 from sperm to zygote might trigger heterochromatin formation in early embryogenesis,

despite the population of H3.3 derived from sperm being diluted during embryogenesis. Furthermore, H3K9me2 at the two paternally imprinted genes, *H19* and *Rasgrf1*, in sperm is transmitted to the zygote, inhibits Tet3-dependent DNA demethylation, and maintains the DNA methylation status by recruiting PGC7[37]. If a subset of H3 in HRCS has the H3K9me2 mark, such nucleosome regions may form silencing chromatin in early embryos by maintaining relatively higher DNA methylation or possibly recruiting gene silencing factors. Although our results indicate that *H19* and *Rasgrf1* are not involved in the H3-binding genes in HRCS (Supplementary Fig. 4d), H3K9me2 signal might be enriched in the small amount of histones on those genes. Similarly, H3K9me2 might be also enriched in the small amount of histone on the category_TS genes, which are suppressed during ZGA in the four-cell embryo. Alternatively, these epigenetic marks in the oocyte (or marks obtained in zygote during early embryonic development) on characterized promoter regions by CpG-enriched sequence may contribute to expressional regulation of these genes.

In the H3 target genes of HRCS, genes involved in neural cell differentiation (pathway of axon guidance and signaling by NGF) are enriched. ES cell differentiation is intrinsically directed to neural cells without external stimulus (the so-called default model), although the molecular mechanism remains unclear[38,39]. This pathway is essentially regulated by zinc-finger nuclear protein Zfp521 with the transcriptional co-activator p300, and artificial expression of Zfp521 can convert ES cells to neural cells even in the presence of BMP4, an antagonistic factor of this pathway[40]. Interestingly, *Zfp521* and *Ep300* (encoding p300) are included in the H3 target genes in HRCS (category_H). One possibility is that, in the early embryo, expression of these neural cell differentiation-related genes is silenced by some epigenetic mark derived from histones in the sperm, which might be important for suppressing the intrinsic capacity of ICM cells for neural differentiation and preventing ectopic differentiation of neural cells in the ICM. Further histone modification analysis in HRCS and in early embryos is required to understand the molecular mechanism of how sperm epigenetic information is transmitted to the embryo.

## Methods

**Preparation of each sperm fraction**. Wild-type male mice on the C57BL/6 background were purchased from Japan SLC and used at 11–12 weeks of age. Sperm cells were collected from caput, corpus, or caudal epididymis or vas deferens. These tissues were incubated in 4 ml of M2 medium (M7167, Sigma) for 1 h at 37 °C, and the whole medium was collected through 70-μm nylon mesh. In the case of preparation of swim-up sperm, the top 2.5 ml or 0.8 ml of medium was collected according to the published protocol[16]. All of the following procedures were performed at 4 °C or on ice. The medium containing each fraction of sperm was centrifuged at $2000 \times g$ for 10 min. Cells ($1.0 \times 10^7$) of sperm pellet were suspended into 1 ml of 50% Percoll and centrifuged at $2500 \times g$ for 5 min without brake to remove the low-density somatic cells, and this step was repeated once. The cell pellet was then suspended into somatic cell lysis buffer (PBS containing 0.1% SDS and 0.5% Triton X-100) and incubated for 10 min, followed by centrifugation at $2000 \times g$ for 3 min. Sperm cell pellets were washed with PBS + BSA (PBS containing 5 mg/ml BSA and 2 mM EDTA) twice and used as total sperm fractions.

To prepare the HRCS fraction, total sperm collected from caudal epididymis and vas deferens was suspended in PBS + BSA ($1.0 \times 10^7$ cells/ml). After mild sonication using handy sonicator (UR-20P, Tomy, Japan) for 10 s at level 1.5 to separate sperm head from the tail, sperm cells were suspended into 1 ml of 82% Percoll and centrifuged at $7700 \times g$ for 5 min without brake to isolate the high-density sperm head fraction, and this centrifugation process was repeated once. After washing the pellet with PBS + BSA twice, sperm cells were used as the HRCS fraction. Experiments were conducted in accordance with the guidelines of the Animal Care and Use Committee of RIKEN Institute.

**SCSA**. The SCSA method was followed as reported previously[12]. Briefly, total sperm and HRCS were suspended in TNE buffer (10 mM Tris-HCl, pH 8.0, 1 mM EDTA, and 0.15 M NaCl) ($1.5 \times 10^6$ cells/ml). Sperm suspension (167 μl) was mixed with 333 μl of acid detergent buffer (0.15 M NaCl and 0.1% Triton X-100, pH 1.2 adjusted by HCl) and incubated on ice for 30 s. Sperm suspension (500 μl)

was neutralized by 1 ml of AO staining buffer (37 mM citric acid, 126 mM Na$_2$HPO4, 1 mM EDTA, 6 μg/ml AO and TO-PRO-3 ( ×1/1000 dilution, Invitrogen), pH 6.0 adjusted by NaOH), and samples were analyzed with an LSRFortessa (BD). The flow cytometry data were processed with FlowJo software (TreeStar). The gating definition was decided by reference to published protocol.

**Western blotting**. Sperm pellet ($1.8 \times 10^6$ cells) was suspended in 180 μl of sperm lysis buffer [PBS containing 0.5% SDS, 10 mM DTT, 1 × complete protease inhibitors (Roche), and 1 mM PMSF]. After incubation for 30 min at 37 °C, sperm lysate was sonicated to shear DNA. For preparation of western blotting samples, 60 μl of 3 × sample buffer was added to 120 μl of lysate. DNA was purified from the remaining 60 μl of lysate, and the DNA amount was used as a loading control. Western blotting samples containing $3–30 \times 10^3$ cells were applied to each well and western blotting analysis was performed. After blocking by PBS containing 3% BSA, the membrane was incubated with H3 antibody (1:3000–10,000 dilution; ab1791, Abcam) or H4 antibody (1:1000 dilution; ab10158, Abcam) and subsequently with peroxidase-conjugated anti-rabbit antibody (1:4000 dilution; Invitrogen). Chemical fluorescence signal was activated by ECL + (PerkinElmer), and the image was scanned with Odyssey systems (LI-COR). Recombinant proteins H3.1 (M2503S, NEB) and H3.3 (M2507S, NEB) were analyzed for checking sensitivities of H3 antibody to H3.1 and H3.3. Uncropped data of western blotting are provided in Supplementary Fig. 12.

**Mass spectrometry**. Sperm pellets (total × 2 and HRCS × 2, both $3 \times 10^6$ cells) were precipitated with TCA and suspended in 1 M Tris-HCl containing 8 M guanidine HCl and 10 mM EDTA, pH 8.5. After reduction with 1,4-dithiothreitol and carboxyl methylation with iodoacetic acid, the samples were precipitated using PAGE cleanup kit (Nacalai tesque, Tokyo, Japan) and digested with trypsin (TPCK treated, Worthington Biochemical Co.) overnight at 37 °C in the buffer of 20 mM Tris-HCl, pH 8.0 containing 0.03% n-dodecyl β-D-maltoside. The digests were analyzed by nano-liquid chromatography–tandem mass spectrometry (MS/MS) using a Q Exactive HFX mass spectrometer (Thermo Fisher Scientific). The peptide mixtures were separated by nano ESI spray column (75 μm [ID] × 100 mm [L], NTCC analytical column C18, 3 μm, Nikkyo Technos) with a gradient of 0–45–90% buffer B (80% (v/v) acetonitrile with 0.1% (v/v) formic acid) in buffer A (MilliQ water with 0.1% (v/v) formic acid) at a flow rate of 300 nL/min over 0–10–30 min (EAST-nLC 1200; Thermo Fisher Scientific).

Mass spectrometer was operated in the positive-ion mode, and the MS/MS spectra were acquired using an inclusion list containing histone H3 variant-specific peptide ions (H3.1: triply charged FQSSAVMALQEA*CEAYLVGLFEDTNL*CAIHAK ion m/z = 1196.89, H3.2: triply charged FQSSAVMALQEASEAYLVGLFEDTNL*CAIHAK ion m/z = 1172.23, H3.3: triply charged FQSAAIGALQEASEAYLVGLFEDTNL*CAIHAK ion m/z = 1146.90, H3t: triply charged FQSSAVMALQEA*CESYLVGLFEDTNL*CAIHAK ion m/z = 1202.23. *C indicate carboxymethyl-cysteine residue). The MS/MS chromatograms of the y14 ion (m/z = 1589.76) of the listed peptides and their MS/MS spectra were drawn using Qual Browser, Thermo Xcalibur 3.1.66.10.

**X-ChIP using sperm cells**. Total sperm and HRCS were cross-linked by 1% formaldehyde for 10 min at room temperature. After quenching with 250 mM glycine, sperm cells were washed with PBS + BSA once and stored at –80 °C. Thawed sperm pellet was washed with sperm decondensation (SpDeC) buffer [5 mM HEPES, pH 8.0, 0.2% NP-40, 10 mM EDTA, 5 mM NaCl, 1.2 M urea, 10 mM DTT, 2 × complete protease inhibitor, and 1 mM PMSF] twice and incubated in SpDeC buffer containing 1 mg/ml heparin sodium salt (H3149, Sigma) for 5 h at 42 °C ($1.5 \times 10^7$ cells of total sperm/3 ml or $1.5 \times 10^7$ cells of HRCS/1.5 ml). Same number ($1.5 \times 10^7$ cells) of total sperm or HRCS was used for each assay.

ChIP experiments after cell lysis step were performed essentially as described[41] with some modifications. Briefly, decondensed sperm cells were washed with lysis buffer 1 (50 mM HEPES, pH 7.5, 140 mM NaCl, 1 mM EDTA, 10% glycerol, 0.5% NP-40, 0.25% Triton X-100, and 1 × complete protease inhibitors) twice and then suspended in elution buffer (50 mM Tris-HCl, pH 8.0, 10 mM EDTA, 1% SDS, 1 × complete protease inhibitors, and 1 mM PMSF). After sonication, lysis buffer 3 (10 mM Tris-HCl, pH 8.0, 100 mM NaCl, 1 mM EDTA, 0.5 mM EGTA, 1% Triton X-100, 0.1% sodium deoxycholate, and 1 × complete protease inhibitors) was added to the sonicated samples and the final SDS concentration was adjusted to 0.1%. After centrifugation for 10 min at $20,000 \times g$, the soluble fractions were pre-cleared by protein A-sepharose beads for 1 h at 4 °C. Input DNA sample was collected from the lysate, and sperm lysate was incubated with the anti-H3 (ab1791, Abcam) antibody-bound protein A-sepharose beads or anti-H3K9me2 (MABI0317, Wako), anti-H3K9me3 (MABI0318, Wako), or mouse IgG (ab18413, Abcam) antibody-bound anti-mouse IgG-conjugated magnetic beads for 14 h at 4 °C. The beads were washed 4 times with ChIP wash buffer (50 mM HEPES, pH 7.0, 0.5 M LiCl, 1 mM EDTA, 0.7% sodium deoxycholate, and 1% NP-40) and twice with TE buffer (10 mM Tris-HCl, pH 8.0, and 1 mM EDTA). The immune complexes were eluted in an elution buffer containing 0.2 mg/ml protamine sulfate to suppress non-specific binding of histones on the tube surface. Note that for preparation of elution buffer,

protamine was added to TE buffer before SDS. The input sample and eluted samples were incubated overnight at 65 °C for reversal of cross-links and then treated with RNase A and proteinase K. DNA was isolated using phenol: chloroform:isoamyl alcohol and ethanol precipitation, and purified with the QIAquick PCR Purification Kit (Qiagen).

Quantitative PCR was performed with a QuantiFast SYBR Green PCR Kit (Qiagen) on a Quant Studio 3 Real-Time PCR System (Applied Biosystems) (primers, Supplementary Table 1). Each quantitative PCR analysis was run with technical triplicates. ChIP efficiency is presented as the percentage of the input sample used for the ChIP lysate.

**Library preparation and sequencing for ChIP-seq analysis**. Input and ChIP'd DNA samples were converted into sequencing library using NEBNext Ultra II DNA Library Prep Kit (New England Biolabs). Samples were end-repaired, adapter ligated, and size-selected using AMPure beads to obtain approximately 200–500 bp final library size. Indexed sequencing libraries were verified by DNA High-Sensitivity chip of Bioanalyzer (Agilent). Paired-end sequencing (2 × 151 cycles) was performed on Illumina NextSeq500 Mid Output Kit v2 (Illumina) at Tsukuba i-Laboratory LLP.

**Processing read data and peak calling**. Paired-end reads were mapped against mouse genome assembly mm10 using CLC Genomics Workbench (v10.1.1, Qiagen) with default settings. Mapped reads were exported as BAM files and converted to BED files by BED tool (v2.27.0) for downstream analysis. Only uniquely mapped reads were analyzed, and proper paired reads were used for peak calling. H3 peak positions were identified using MACS2 at FDR < 0.01 and FC > 5.

**Genome coordinates**. Genomic compartments were defined with ncbiRefSeq data obtained by UCSC table browser. The features were defined as follows: promoters (1 kb around the annotated TSS), exons, introns, upstream regions (from 5 kb upstream of the TSS), and downstream regions (from 5 kb downstream of the TES). Intergenic regions were classified into intergenic repeat and intergenic unique regions by overlapping of repeat regions, defined with rmsk data obtained using UCSC table browser.

**Quantification of o/e ratio in CpG**. Genomic positions of CpG islands were defined with cpgIslandExt data obtained using UCSC table browser. Values of o/e ratios included in the above data were calculated.

**Targeted bisulfite sequencing (TGBS)**. Targeted bisulfite sequencing (TGBS) libraries were prepared according to the post-bisulfite adapter tagging (PBAT) protocol with slight modifications[42,43]. Briefly, 300 ng of genomic DNA was fragmented to ~500 bp with Covaris S220 and used for target enrichment with SureSelectXT Mouse Methyl-Seq kit (Agilent). The enriched DNA was subjected to PBAT, followed by 5 cycles of PCR enrichment with Kapa Library Amplification Kit (Illumina). Single-read sequencing (100 cycles) was performed on Illumina HiSeq 2500 using HiSeq SR Rapid Cluster Kit v2 (Illumina) and HiSeq Rapid SBS Kit v2 (Illumina). Two indexed libraries were combined with a high concentration of spike-in PhiX control DNA (20%) and loaded on a single lane of the flow cell.

**TGBS data analysis**. TGBS reads were mapped to the mouse reference genome sequence (mm9) using Bmap (http://itolab.med.kyushu-u.ac.jp/BMap/index.html). Methylation levels were calculated for individual CG sites. Methylation levels of probe regions and H3 peak regions were determined by averaging the methylation levels of CG sites in individual features. The coordinates of H3 peak regions on mm10 assembly were converted to those on mm9 using the UCSC LiftOver tool for the integrative analyses of DNA methylation and histone deposition. This process filtered out seven out of 1320 H3 peak regions in HRCS and 41 out of 10,998 regions in total sperm. Correlation of methylation levels between HRCS and swim-up sperm was evaluated by calculating Pearson's correlation coefficient and displayed as a scatter plot.

**Comparison of our data with published datasets**. Published ChIP-seq, ATAC-seq, and RNA-seq data were obtained from NCBI GEO to compare them with our results, and query codes of the datasets used are summarized in Supplementary Table 2.

**Preparation of profile plot data and heatmap data**. The number of raw reads mapped on 3 kb around the TSS was counted, and read coverage was calculated after normalization by mapped read number. Matrix data of read coverage against each target promoter region in 50-bp bin size were prepared using deepTools2[44]. Using this matrix, profile plots were prepared using average values for target regions. The degree of read coverage in each target promoter region is indicated by the heatmap.

**Gene functional analysis**. Functional analysis of H3 target genes was performed by PANTHER classification system[45]. Using 820 category_H and 5608 category_

TS target genes, which are found in PANTHER database respectively, enrichment terms of Reactome pathway and GO simple biological process were identified.

**Statistics**. For SCSAs, mean ± s.d. is shown in bar graph with corresponding dot plots to observed values. In all box plot, elements indicate the following values; center line, median; box limits, upper and lower quartiles 1.5x interquartile range; and points, outliers. The significance of difference in o/e ratio, DNA methylation, and H3K4me3 levels was evaluated with the Wilcoxon rank-sum test. A sample size in this study was decided on the basis of past experience in generating statistical significance. Investigators were not blinded to experimental conditions, and no randomization or exclusion of data points were used.

## Data availability

H3 ChIP-seq data and TGBS data are deposited in the Gene Expression Omnibus (GEO) under accession code GSE113150.

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

## Acknowledgements

We thank K. Mochida and A. Ogura (RIKEN) for showing the unpublished data and useful discussion; K. Yamaguchi and Y. Okada (University of Tokyo) for useful discussions. This work was supported in part by a Grant-in-Aid for Scientific Research on Innovative Areas from the Ministry of Education, Culture, Sports, Science, and Technology of Japan, and by AMED under Grant Numbers JP16am0101059, JP16am0101060, JP18am0101103, and JP18am0101105.

## Author contributions

K.Y. performed SCSA and immunoblotting analysis for each sperm sample, and did H3 ChIP-seq experiment. T.S. and N.D. performed quantitative protein analysis by LC-MS. M.M., R.N., and K.S. analyzed the data from ChIP-seq experiments. F.M., H.A., and T.I. analyzed the data from bisulfite-seq experiments. S.I. conceived and supervised the whole study, and S.I. and K.Y. wrote the paper.

## Additional information

**Competing interests:** M.M. declares association with Tsukuba i-Laboratory as a technical consultant.

