## [Peer Review File · Nature Communications]

Reviewers' comments:

Reviewer #1 (Remarks to the Author):

As highlighted by the authors in their manuscript, conflicting reports on histone retention in mature spermatozoa (mammalian) exist in the literature.

Here the authors propose a new approach to the mapping of retained histones that is based on prior rigorous characterization of sperm cells. They convincingly show that sperm populations used in most of the previously reported studies contain a fraction of cells that has not completed the histone-to-protamine replacement.

The authors report a new protocol for mature sperm cell purification that strongly reduces the "immature" cell fraction. Additionally, they manage to define a new protocol for sperm genome decompaction and the isolation of histone-containing fraction of the genome without the use of MNase digestion.

Using these innovative approaches they establish a high confidence map of H3-associated genome regions and proceed with the characterization of these regions in terms of sequence characteristics, DNA methylation and regulatory elements. They then exploit published data on preimplantation embryos to highlight the association between their findings on gene-associated histone retention in sperm cells and the behaviour of the corresponding genes in early embryos. Overall this is an important contribution to the field and could be of interest to a large audience interested in epigenetics.

However, before publication the authors need at least to discuss one important point, which deals with the real nature of the histone-associated genome regions in mature spermatozoa.

A recently published work showed the nature of transitional structures that appear during the histone-to-Prm replacement, where histone and non-histone proteins could co-exist. These structures released by MNase, are associated with shorter MNase-resistant DNA fragments, compared to regular nucleosomes (PMID:28366643). These structures also seem to persist in mature spermatozoa (PMID:24998598 ; PMID:28366643).

However, in mature sperm the ratio of regular nucleosomes and sub-nucleosomes is not known. In the previous work based on MNase digestion the authors mostly look at regular nucleosomes, while here the authors' protocol does not allow to distinguish between regular and sub-nucleosomal fractions of the genome.

Indeed, the sub-nucleosomal fragments are less resistant to MNase digestion and could disappear after extensive MNase digestion (PMID:17261847), and hence may be absent in the published mapping data using MNase (except if the authors pay attention to the shorter fragments, i. e., PMID:24998598).

Here, the authors should therefore be aware that they are probably dealing with both nucleosomal and non-nucleosomal histones.

These could contribute to some differences observed between the data reported here and the previously published mapping data.

To avoid any over-interpretation of the data presented here, it is crucial that the authors develop a deep discussion on the mechanism of histone-to-Prm replacement and the possibility of the retention of non-nucleosomal histone-containing structures, as well as the contribution of such structures in their mapping data.

Reviewer #2 (Remarks to the Author):

The paper by Yoshida et al. employs a novel method of purifying "mature sperm" that have completed the histone-to-protamine exchange (HRCS). This histone to protamine exchange process continues as sperm samples progress through the epididymis. To determine where histones are retained in HRCS samples, and remove contaminating signals from supposedly

immature cells, the authors performed H3 chip-seq analysis on the HRCS and total sperm samples. Interestingly, the H3-ChipSeq data from total and from HRCS sperm samples confirm that histones are retained at developmental gene promoters and CpG islands as previously described (Erkek et al.). This paper adds an important contribution to the field given the debate regarding nucleosome retention and localization in sperm. Further, given methodological concerns regarding MNase over-digestion, the use of sonication and complete histone recovery adds important findings to the field.

Major Concerns:

1) The authors developed a sperm separation technique to distinguish between compact mature sperm heads from immature sperm. Is there any clinical/translational significance to this population isolated using this novel method? Do these sperm heads actually have higher fertilization potential, better rates of embryogenesis or improved pregnancy outcomes?

2) The authors compare histone localization in HRCS sample vs. total sperm. H3 signals in both total sperm and HRCS sperm were mainly found on gene promoters, however, the signals were less intense or absent in HRCS sperm. Was a similar number of sperm used for both IPs. ChipSeq is not quantitative – to discuss the relative differences in enrichment the authors will need to calibrate their chip-seq experiments with mononucleosomal standards developed by the Ruthenburg lab.

3) Second, the overlap between the HRCS and total was about 70%. However, it was really surprising to see a 20% overlap between HRCS and previous work by Erkek et al. how do the authors reconcile the difference? What genes/pathways are shared or distinct? Do the authors see better correlations when comparing the H3 data from total and HRCS to the recent work by Jung et. al. Cell reports 2017. This sample was prepped by sonication rather than MNase.

4) Furthermore, it would be more interesting to compare the HRCS fraction to the more immature fraction rather than total. Do you see paring down of the histone domains in HRCS samples consistent with overall reduction in histone retention?

5) In figures 4 – the author reprocessed previously published bulk h3k4me3 and atac-seq data to glean insights on epigenetic modifications and chromatin state of HRCS histones. IPs should be performed on HRCS samples. By intersecting the two datasets you can't determine whether the HRCS histones actually bear these modifications.

Minor comments:

1) Mass spec pie chart for the total sperm is above 100. There can be a rounding error. Also please explain how your mass spec quantification of the histone H3 variants was performed between the groups in the methods.

2) Define PPT in Figure 2? Why is there no DNA detected in pellet – I am assuming the pellet includes the protamine associated DNA.

3) Figure 3 Define WCE? Is this the total input DNA?

4) In Page 4-5 the authors mention : " The DNA fragmentation index...the content of DFI fraction in mouse sperm samples is usually lower than in human samples, because mouse sperm can be analyzed immediately after preparation" Is this actually true? What about the fact that mouse sperm has less histone overall than human, I would imagine that this also contributes.

5)The use of "immature sperm" throughout the paper is confusing. Unless the sperm is morphologically assessed to be immature, I am not sure you can call them immature. They may have faulty histone-to-protamine exchange which may be an aberration instead of a marker of immaturity. Do the sperm from the different sections of the epididymis have morphological features that distinguish them?

6)Page 7 " to examine the quality of sperm, we first measure population with the degree of nuclear condensation..." the wording is confusing and I am not sure what they authors are trying to say.

7)Page 9: "These results also suggest that about 80% of H3 protein contained in total and swim-up sperm samples is derived from immature sperm..." I do not follow where this conclusion is drawn from.

8)What is the difference in the results of the SCSA using the percoll isolation of HRCS if the tails are not first sonication? In other words why do the authors think that sonication of the tails followed by the percoll has such an impact on isolation of such a pure population? How is this different from just using Percoll?

Reviewer #3 (Remarks to the Author):

Most of the DNA in mature sperm is highly condensed by protamines, which replace almost all of the histones. However, a very small number of nucleosomes (~0.3%) are retained. Surprisingly, the persistent nucleosomes are not randomly located with respect to the genome. This paper addresses the controversial question of the locations of residual nucleosomes in mouse and human sperm. Previous studies disagree over which regions of the genome are enriched in nucleosomes: some argue that the nucleosomes are enriched at some promoters, while others argue that they are enriched in distal intergenic regions and at repeated sequences. These studies are based mostly on MNase-seq and there is some concern in the literature over whether the extent of chromatin digestion by micrococcal nuclease is responsible for the disagreement. In my opinion, it is difficult to see how the extent of MNase digestion could affect nucleosome locations in such a gross way; the problem must lie elsewhere. In any case, the authors have resolved the controversy by developing methods for removing immature sperm from the preparation and for reliable ChIP-seq from sperm using sonication to avoid MNase altogether. Removal of immature sperm is important because they have higher histone content than mature sperm and may bias the data. The new method, using Percoll gradient separation instead of the "swim-up sperm" method, may have clinical relevance in selecting the most viable sperm for in vitro fertilisation. The ChIP-seq method is important because sperm DNA is difficult to disperse and sonicate. The authors find that sperm nucleosomes are enriched primarily at promoters. Overall, this is a nice paper, clearly written and the data support the claims.

I have the following comments:

1. In Fig. 3b, the distributions of histone H3 peaks in different regions of the total sperm and purified mature sperm (HRCS) genomes are shown as pie charts. Do these distributions take into account the relative sizes of the regions involved? I don't think they do. If not, it might be worth pointing out that 64% of the peaks are at promoters even though promoters account for a very small part of the genome. What about H3 peaks at enhancers?

2. On page 13, it is stated that, referring to Fig. 4a: "At the 4-cell stage, category_H and category_PH genes showed lower ATAC-seq and H3K4me3 ChIP-seq signals than category_TS

genes". However, although this is true for the ATAC-seq data, it is not very clear for the H3-K4me3 data (Fig. 4a).

3. On page 14 and in Fig. 4, the authors use data for early embryos to try to address whether the nucleosomes at promoters in the mature sperm affect gene expression during subsequent development. It is argued that more nucleosome-marked mature sperm genes are highly expressed than the average in oocytes (Fig. 4d). This argument seems weak. It might be better to separate the category H genes into those that are expressed in oocytes and those that are not, and then examine the characteristics of these two groups of genes as the embryo develops. Also, the authors do not discuss the possible contribution of the chromatin structure of the oocyte genome.

Minor points:

4. I suggest retention of Reference Figure 1 - it is very helpful.

5. There are some typographical errors in the figures: "category", "principal" instead of "principle", "immaure", "damege" etc.

6. In the Abstract and elsewhere, the authors state that their method results in completely solubilised nucleosomes. However, a nucleosome would not survive the 1% SDS step in their method, unless all of the histones are cross-linked to DNA, which is unlikely to be the case (under the conditions used, most histones would not be cross-linked to DNA).

Reply to Reviewers

Please see Reviewers' comments below in **blue** and also our reply in **black**.

Reviewer #1 (Remarks to the Author):

However, before publication the authors need at least to discuss one important point, which deals with the real nature of the histone-associated genome regions in mature spermatozoa.

A recently published work showed the nature of transitional structures that appear during the histone-to-Prm replacement, where histone and non-histone proteins could co-exist. These structures released by MNase, are associated with shorter MNase-resistant DNA fragments, compared to regular nucleosomes (PMID:28366643). These structures also seem to persist in mature spermatozoa (PMID:24998598 ; PMID:28366643).

However, in mature sperm the ratio of regular nucleosomes and sub-nucleosomes is not known. In the previous work based on MNase digestion the authors mostly look at regular nucleosomes, while here the authors' protocol does not allow to distinguish between regular and sub-nucleosomal fractions of the genome.

Indeed, the sub-nucleosomal fragments are less resistant to MNase digestion and could disappear after extensive MNase digestion (PMID:17261847), and hence may be absent in the published mapping data using MNase (except if the authors pay attention to the shorter fragments, i. e., PMID:24998598).

Here, the authors should therefore be aware that they are probably dealing with both nucleosomal and non-nucleosomal histones.

These could contribute to some differences observed between the data reported here and the previously published mapping data.

To avoid any over-interpretation of the data presented here, it is crucial that the authors develop a deep discussion on the mechanism of histone-to-Prm replacement and the possibility of the retention of non-nucleosomal histone-containing structures, as well as the contribution of such structures in their mapping data.

Reply:

Thank you for your useful and important comment. Based on your comment, we have added one paragraph in the Discussion as shown below to discuss the mechanism of histone-to-Prm replacement and the possibility of the retention of non-nucleosomal histone-containing structures, as well as the contribution of such structures in our mapping data.

Page 15, line 18:

Recently, Barral *et al.* analyzed the process of the histone-to-protamine replacement and showed the nature of transitional structures^{xx}. In condensing spermatids, the H2A.L.2-TH2B dimer is first loaded onto the nucleosomes, which opens the nucleosomes and allows the invasion of nucleosomes by transition

proteins. Nucleosome-transition proteins interact with incoming protamines, which mediate pre-protamine-2 processing, and protamines then bind DNA in competition with histone-transition protein complexes. Since the displaced histones are unable to remain as octamers, protamine-DNA and displaced transition protein-histone complexes constitute a relatively stable transitional state, generating the small subnucleosomal structures. HRCS might involve the small subnucleosomal structures, and our sample may contain them because MNase was not used in our method to discriminate the normal nucleosomal structures and the small subnucleosomal structures. Thus, further analyses might be needed to examine whether the small subnucleosomal structures might contribute to some H3-binding genes detected in HRCS.

Reference:

Barral, S. *et al.* Histone variant H2A.L.2 guides transition protein-dependent protamine assembly in male germ cells. *Mol. Cell* **66**, 89-101 (2017).

Reviewer #2 (Remarks to the Author):

Major Concerns:

1) The authors developed a sperm separation technique to distinguish between compact mature sperm heads from immature sperm. Is there any clinical/translational significance to this population isolated using this novel method? Do these sperm heads actually have higher fertilization potential, better rates of embryogenesis or improved pregnancy outcomes?

Reply:

Thank you for your useful questions. At present, we have no clinical and translational significance to the population isolated using this new method. Apparently, this is important future work, especially in the applied field.

However, some previous studies strongly suggest that the sperm heads prepared by this novel method have higher fertilization potential. Our novel method succeeded to remove HDS sperm fraction, and it was reported that the group with a larger proportion of HDS sperm in humans has a lower chance of pregnancy (refs. 13, 14 in the original manuscript).

Furthermore, it was reported using various mammals that the quality of sperm prepared by Percoll-dependent method is better than that by swim-up method (for example, Arias, M.E. *et al.* *Reprod. Bio.* (2017)). Thus, we speculate that fertilization potential of HRCS fraction may be significantly higher than total or swim-up sperm fraction which contains significant population of more histone-residual sperm.

Reference:

Arias, M.E. *et al.* Bovine sperm separation by Swim-up and density gradients

(Percoll and BoviPure): Effect on sperm quality, function and gene expression. *Reprod Biol.* **17**, 126-132 (2017).

Sakkas, D. *et al.* The use of two density gradient centrifugation techniques and the swim-up method to separate spermatozoa with chromatin and nuclear DNA anomalies. *Hum Reprod.* **15**, 1112-1116 (2000).

2) The authors compare histone localization in HCRS sample vs. total sperm. H3 signals in both total sperm and HCRS sperm were mainly found on gene promoters, however, the signals were less intense or absent in HCRS sperm. Was a similar number of sperm used for both IPs. ChipSeq is not quantitative – to discuss the relative differences in enrichment the authors will need to calibrate their chip-seq experiments with mononucleosomal standards developed by the Ruthenburg lab.

Reply:

We used the same number of sperm cell and the same amount/lot of antibody for each ChIP-seq experiment using HRCS sample or total sperm (1.5×10^7 cells/6 μ g antibody/assay). As described in the paper which reviewer kindly introduced, ChIP-seq analysis is not quantitative method, and IceChIP method was developed for quantification of epigenetic marks in genome-wide level. However, this method has been recently used for the comparison of level of different histone modifications (for example, Werner M.S. *et al.* NSMB (2017)). In the case of ChIP experiments using the same antibody, this method has not been used, and many scientists performed the experiments as we did. Some H3-binding sites both of which have similar level of H3 signal in total showed significantly different level of signal of H3 signal in HRCS (**Suppl. Fig. 4c** in the revised manuscript). This also supports the notion that reduction of H3 signal in some regions in HRCS is not due to a different condition of ChIP-seq.

Reference:

Werner M.S. *et al.* Chromatin-enriched lncRNAs can act as cell-type specific activators of proximal gene transcription. *Nat. Struct. Mol. Biol.* **24**, 596-603 (2017).

Hattori T. *et al.* Antigen clasping by two antigen-binding sites of an exceptionally specific antibody for histone methylation. *Proc Natl Acad Sci U S A.* **113**, 2092-7 (2016).

To clarify experimental condition, following statement was added in method section for X-ChIP using sperm cells.

Page 29, line 19:

“Same number (1.5×10^7 cells) of total sperm or HRCS was used for each assay”

3) Second, the overlap between the HCRS and total was about 70%. However, it was really surprising to see a 20% overlap between HCRS and previous work by

Erkek *et al.* how do the authors reconcile the difference? What genes/pathways are shared or distinct? Do the authors see better correlations when comparing the H3 data from total and HRCS to the recent work by Jung *et al.* Cell reports 2017. This sample was prepped by sonication rather than MNase.

Reply:

Poor overlapping between Erkek's and our data may be caused by both the sperm purification method and the mapping method. Erkek *et al.* used swim-up sperm population which still contains the histone-replacement uncompleted sperm (about 10% of total population), while we used HRCS lacking the histone-replacement uncompleted sperm. Because the swim-up sample involves more amount of H3, this might cause more number of detected peaks compared to HRCS. While Erkek *et al.* identified the nucleosomal regions by MNase-seq, we did H3 binding sites by ChIP-seq. This difference also produced some bias to experimental results.

What genes/pathways are shared or distinct?

We also performed pathway analysis for the genes identified by Erkek *et al.* of which promoters contain nucleosome (**Suppl Fig. 8a** in the revised manuscript). Four pathway terms of 5 top terms in this result are overlapped with them in Category_TS, and "Developmental Biology" in Category_H. Thus, only the "development-related genes" are shared between our results and Erkek's data, while other pathway-related genes identified by Erkek *et al.* were absent in HRCS. To clearly describe this, we have added the following statement.

Page 13, line 13:

Only development-related genes were shared between category_H genes and nucleosome target genes previously identified by MNase-seq¹⁶ (**Supplementary Fig. 8a**).

In the Cell Reports paper by Jung *et al.* (2017), ChIP experiments were performed only for CTCF and Smc1, but not for histones.

4) Furthermore, it would be more interesting to compare the HRCS fraction to the more immature fraction rather than total. Do you see paring down of the histone domains in HCRS samples consistent with overall reduction in histone retention?

Reply:

I agree your suggestion that direct comparison between HRCS and immature sperm fractions is interesting. To this end, we need to prepare pure immature sperm fraction. Actually, we tried to purify immature sperm from supernatant fraction after 82% Percoll purification, but it also contained the tail-retaining mature sperm which survived from mild sonication. In our protocol for HRCS purification, very mild sonication (only 10 sec at minimum output level) was used for preparation of sperm head to avoid some damage to sperm head. Some additional technical trick is required to prepare pure fraction of immature sperm.

However it is not major point in this paper, so that we used total sperm fraction.

In western blotting result using HRCS, we did not detect any cleaved band of histone (**Supplementary Fig. 12** in the revised manuscript). This indicates that the amount of full histone protein is reduced in HRCS compared to the total sperm.

5) In figures 4 – the author reprocessed previously published bulk h3k4me3 and atac-seq data to glean insights on epigenetic modifications and chromatin state of HCRS histones. IPs should be performed on HCRS samples. By intersecting the two datasets you can't determine whether the HRCS histones actually bear these modifications.

Reply:

Thank you for your useful comment. Our results suggest that the histone binding genes in HRCS may be suppressed at the stages between 4-cell and ICM (**Suppl. Fig. 11a** in the revised manuscript). To test whether this might be correlated with the presence of epigenetic silencing marks in HRCS, we have examined whether these genes have the histone H3K9me2/3 marks, as recommended by the Reviewer. Results of q-ChIP indicate that these genes have the H3K9me2 mark (**Supple Fig. 11b** in the revised manuscript). We have described these results as follows.

Page 14, line 14:

To test whether this might be correlated with the presence of epigenetic silencing marks in HRCS, we have examined that these genes have the histone H3K9me2/3 marks. Actually, H3K9me2, not H3K9me3, was found in the promoter regions of category_H genes which expressions were suppressed at the stages between 4-cell and ICM (**Supplementary Fig. 11a, b**).

Minor comments:

1) Mass spec pie chart for the total sperm is above 100. There can be a rounding error. Also please explain how your mass spec quantification of the histone H3 variants was performed between the groups in the methods.

Reply:

We checked original data of Mass spec., and found a rounding error. To avoid this, percentage in pie chart was shown to two places of decimals (**Supplementary Fig. 3e** in the revised manuscript).

For mass spec quantification, we draw MS/MS chromatogram of histone H3 variant specific peptide ions in **Suppl. Fig. 3** (a ; H3.1, b ; H3.2, c ; H3.3, d ; H3t) as mentioned in the methods. The quantification of the histone H3 variants was performed using areas of these chromatograms.

2) Define PPT in Figure 2? Why is there no DNA detected in pellet – I am assuming the pellet includes the protamine associated DNA.

Reply:

PPT means “cell pellet” after sonication and centrifugation for sperm lysate. We succeeded complete solubilization of sperm nuclear DNA/protein including nucleoprotamine and nucleosome by decondensation treatment of sperm nuclear (**Fig. 2a**). Actually, almost no DNA was detected by gel electrophoresis in DNA sample prepared from pellet fraction (**Fig. 2c**).

3) Figure 3 Define WCE? Is this the total input DNA?

Reply:

Yes. WCE (Whole Cell Extract) means “total input DNA” which corresponds to sperm lysate after pre-clear beads treatment. To avoid confusing of this term, “WCE” was replaced by “input”.

4) In Page 4-5 the authors mention: “ The DNA fragmentation index...the content of DFI fraction in mouse sperm samples is usually lower than in human samples, because mouse sperm can be analyzed immediately after preparation” Is this actually true? What about the fact that mouse sperm has less histone overall than human, I would imagine that this also contributes.

Reply:

Thank you for your comment. We agree with your opinion that less histones in mouse sperm may also contribute to lower content of DFI fraction compared to that in human sperm. Therefore, we have revised to following.

Page 4, line 21:

The content of DFI fraction in mouse sperm samples is usually lower than in human samples, possibly because mouse sperm can be analyzed immediately after preparation and have less histones compared to human sperm.

5) The use of “immature sperm” throughout the paper is confusing. Unless the sperm is morphologically assessed to be immature, I am not sure you can call them immature. They may have faulty histone-to-protamine exchange which may be an aberration instead of a marker of immaturity. Do the sperm from the different sections of the epididymis have morphological features that distinguish them?

Reply:

Thank you for your comment. To avoid confusing wording in our manuscript, term of “immature sperm” in some points has been replaced by “sperm which have not yet completed the histone-to-protamine replacement” or “histone replacement-uncompleted sperm (HRunCS)”.

6) Page 7 “ to examine the quality of sperm, we first measure population with the degree of nuclear condensation...” the wording is confusing and I am not sure what they authors are trying to say.

Reply:

We are sorry for confusing wording. To clear up what we want to say, we have revised to following.

Page 7, line 2:

To examine the quality of sperm, we first analyzed sperm using the SCSA¹².

7) Page 9: “These results also suggest that about 80% of H3 protein contained in total and swim-up sperm samples is derived from immature sperm...” I do not follow where this conclusion is drawn from.

Reply:

We are sorry for confusing wording. To clear up what we want to say, we revised to following.

Page 9, line 5:

These results suggest that HRunCS contains much more amount of histone proteins than HRCS. In total sperm, 10% of them was HRunCS, and results of western blotting indicated that the ratio of H3 in total sperm (90% HRCS + 10% HRunCS) and HRCS (100% HRCS) is 5 : 1. These results indicate that H3 protein contained in total and swim-up sperm samples is mainly derived from HRunCS and that the distribution of histone-binding sites in the swim-up sperm reflects the localization of histones in HRunCS.

8) What is the difference in the results of the SCSA using the percoll isolation of HRCS if the tails are not first sonication? In other words why do the authors think that sonication of the tails followed by the percoll has such an impact on isolation of such a pure population? How is this different from just using Percoll?

Reply:

We did not try the Percoll isolation of HRCS without sonication, because this appeared to be difficult based on the following observation. Only small fraction of the sperm complex consisted of head and tail were precipitated through 60% Percoll, while all the sperm head were precipitated through 90% Percoll. This indicates that the sperm tail has much lower specific gravity than sperm head, and may disturb the Percoll separation of mature and immature sperm. Therefore, we first removed the tail by mild sonication.

Reviewer #3 (Remarks to the Author):

1. In Fig. 3b, the distributions of histone H3 peaks in different regions of the total sperm and purified mature sperm (HRCS) genomes are shown as pie charts. Do these distributions take into account the relative sizes of the regions involved? I don't think they do. If not, it might be worth pointing out that 64% of the peaks are at promoters even though promoters account for a very small part of the genome. What about H3 peaks at enhancers?

Reply:

Thank you for your useful comments. Population of genomic background in each functional region is different, and so we show enrichment of peak numbers in each genomic region after normalization of genomic background (**Suppl. Fig. 5a** in the revised manuscript). As you suggested, H3 peaks are highly enriched in promoter region at about 30 fold relative to genomic background.

We also found around 5% of H3 peaks in HRCS, relatively minor population of whole peaks was overlapped with enhancer regions reported by following paper.

Reference:

Shen Y. *et al.* A map of the cis-regulatory sequences in the mouse genome. *Nature*. 488, 116-20 (2012).

2. On page 13, it is stated that, referring to Fig. 4a: "At the 4-cell stage, category_H and category_PH genes showed lower ATAC-seq and H3K4me3 ChIP-seq signals than category_TS genes". However, although this is true for the ATAC-seq data, it is not very clear for the H3-K4me3 data (Fig. 4a).

Reply:

To more clearly indicate the difference, we have also prepared box-plot figure for H3K4me3 ChIP-seq coverage for category_H, category_PH and category_TS, and put it (**Suppl. Fig. 8b** in the revised manuscript).

3. On page 14 and in Fig. 4, the authors use data for early embryos to try to address whether the nucleosomes at promoters in the mature sperm affect gene expression during subsequent development. It is argued that more nucleosome-marked mature sperm genes are highly expressed than the average in oocytes (Fig. 4d). This argument seems weak. It might be better to separate the category H genes into those that are expressed in oocytes and those that are not, and then examine the characteristics of these two groups of genes as the embryo develops. Also, the authors do not discuss the possible contribution of the chromatin structure of the oocyte genome.

Reply:

Thank you for your kind suggestion. We recalculated population number of highly expressed genes after separation by expression level in oocyte (FPKM>10). Actually, we found that category_H and category_PH genes clearly show different pattern of expression from other two groups, and Fig. 4d has been replaced by new one. Thank you for your useful comment!

We have added following sentence for possible contribution of the chromatin structure of the oocyte genome in the discussion part.

Page 18, line 10:

Alternatively, these epigenetic marks in oocyte (or mark obtained in zygote during

early embryonic development) on characterized promoter regions by CpG enriched sequence may contribute to expressional regulation of these genes.

Minor points:

4. I suggest retention of Reference Figure 1 - it is very helpful.

Reply:

Thank you for your kind comment. Since the original Reference Figure 1 was taken from the previously published paper, we have remade similar Figure, and added as **Supplementary Fig. 1**.

5. There are some typographical errors in the figures: "category", "principal" instead of "principle", "immaure", "damege" etc.

Reply:

We are sorry for these typographical errors. We have corrected them.

6. In the Abstract and elsewhere, the authors state that their method results in completely solubilised nucleosomes. However, a nucleosome would not survive the 1% SDS step in their method, unless all of the histones are cross-linked to DNA, which is unlikely to be the case (under the conditions used, most histones would not be cross-linked to DNA).

Reply:

We agree your comment for possibility that some population of histone is released from DNA after sonication treatment. To avoid the overstatement, we have revised to following.

“to completely solubilize nucleosome” → “to completely solubilize histones”

REVIEWERS' COMMENTS:

Reviewer #1 (Remarks to the Author):

None

Reviewer #3 (Remarks to the Author):

The revised manuscript addresses my previous concerns.